# Prevalence of Wheezing and Its Association with Environmental Tobacco Smoke Exposure among Rural and Urban Preschool Children in Mpumalanga Province, South Africa

**DOI:** 10.3390/ijerph21040469

**Published:** 2024-04-11

**Authors:** Rodney Mudau, Kuku Voyi, Joyce Shirinde

**Affiliations:** 1Department of Human Nutrition, Faculty of Health Sciences, University of Pretoria, Private Bag X323, South Africa; 2School of Health Systems and Public Health, Faculty of Health Sciences, University of Pretoria, Private Bag X323, South Africa; kuku.voyi@up.ac.za (K.V.); joyce.shirinde@up.ac.za (J.S.)

**Keywords:** environmental tobacco smoke exposure, the risk of wheezing, poor air quality, preschool children

## Abstract

Background: This study aimed to investigate the prevalence of wheezing and its association with environmental tobacco smoke exposure among rural and urban preschool children in Mpumalanga province, South Africa, an area associated with poor air quality. Methods: In this study, parents/caregivers of preschool children (*n* = 3145) completed a modified International Study of Asthma and Allergies in Childhood (ISAAC) questionnaire. Data were analysed using multiple logistic regression models. Results: The overall prevalence of Wheeze Ever was 15.14%, with a higher prevalence in urban preschoolers than rural preschoolers (20.71% vs. 13.30%, *p* < 0.000). Moreover, the total prevalence of Asthma Ever was 2.34%. The prevalence was greater in urban preschoolers than in rural preschoolers (3.92% vs. 1.81%, *p* < 0.001). In the final adjusted model, both urban- and rural-area children who lived with one or more people who smoked in the same household (WE: OR 1.44, 95% CI 1.11–1.86) (CW: OR 2.09, 95% CI 1.38–3.16) and (AE: OR 2.49, 95% CI 1.12–5.54) were found to have an increased likelihood of having Wheeze Ever, Current Wheeze, and Asthma Ever as compared to those who lived with non-smokers. Conclusions: The implementation of smoking limits and prohibition is crucial in areas that are frequented or utilized by children. Hence, it is imperative for healthcare providers to actively champion the rights of those who do not smoke within the society, while also endorsing legislative measures aimed at curtailing the extent of tobacco smoke exposure.

## 1. Introduction

Worldwide, there are differences in the prevalence of preschool wheezing, and it appears to be rising [1]. Wheeze can be described as a persistent high-pitched sound characterized by a melodic tone that originates from the chest during the act of exhaling [1,2]. Wheezing throughout early childhood is a prevalent yet intricate symptom characterized by multiple aetiologies and potential consequences [2]. Moreover, wheezing in preschoolers results in significant healthcare costs and unscheduled hospital consultations [3].

It is common for children who exhibit wheezing symptoms before the age of three, and persist with wheezing until the age of six, to possess atopic tendencies and subsequently develop asthma over the years [4,5]. Moreover, it has been observed that the respiratory capacity of children experiencing wheezing tends to enhance as they grow older; however, their respiratory capacity never reaches the level observed in children who have never experienced wheezing [4].

Preschool wheezing is a common condition, and environmental tobacco smoke (ETS) exposure is a significant risk factor for wheezing in preschool children [6,7,8]. ETS possess comparable toxic components to those found in conventional tobacco smoke, consequently resulting in similar detrimental consequences akin to those observed in individuals who engage in active smoking [9]. The symptoms of wheezing may exhibit temporary remission following therapy interventions and/or the avoidance of triggers associated with the condition. Hence, it is advisable to enact legislation aimed at the elimination and regulation of children’s exposure to ETS.

According to the Tobacco Products Control Act of 1993 in South Africa, specifically Section 2(1)(a)(iii), it is prohibited for individuals to engage in smoking any tobacco product within a motor vehicle in the presence of a child under the age of 12 years. This provision has been subject to amendments. The act of smoking is now prohibited within buildings designated for commercial childcare services. Sweets and toys resembling cigarettes are likewise prohibited. The implementation of new tobacco regulations was officially announced in September 2022 through the publication of Government Gazette Staats koerant [10].

The following are highlights that will be considered regarding the risk of childhood exposure to ETS:

In the event that a residence is utilized for educational purposes, tutoring services, or commercial childcare, the act of smoking is prohibited.There is implementation of a prohibition on smoking within motor vehicles in the presence of a minor under the age of 18, provided that there is more than one individual occupying said vehicle.The proposed legislative expansion involves not only traditional cigarettes, but also embraces any devices used in connection with tobacco-related goods and electronic delivery systems, such as pipes, water pipes, and electronic devices.

There is a tendency for wheezing prevalence to be lower in rural areas, with some evidence suggesting the presence of an urban–rural gradient [11,12]. The urban–rural gradient of wheeze in preschool children is examined in this study to determine whether this indicator differs along an urban–rural gradient.

The current tobacco control laws are introduced as a baseline and the impact of the regulations will be seen in later years. This study presents the baseline of the prevalence of wheezing in Mpumalanga where children are exposed to polluted air including ETS. The aim was to assess the prevalence of wheezing and its association with environmental tobacco smoke exposure among rural and urban preschool children in Mpumalanga province, South Africa.

## 2. Methods

### 2.1. Study Design and Settings

An analytical cross-sectional survey was conducted between November 2020 and April 2021. The objective of our study was to assess the prevalence of wheezing and its association with environmental tobacco smoke exposure, a common symptom of asthma, among preschool children residing in rural and urban areas. The research was carried out within the Mpumalanga province, specifically in the Gert Sibande district municipality, which is situated within the Highveld Priority Area. In accordance with the National Environmental Management: Air Quality Act, 2004 (Act No. 39 of 2004), the Minister of Environmental Affairs named this region a priority area for air pollution in 2007.

The Gert Sibande district municipality was purposively selected because it is in the Highveld priority area. The Highveld priority area has substandard air quality and heightened levels of pollutants originating from both industrial and non-industrial origins. The district encompasses a diverse range of sectors, such as power generating, petrochemical, primary metallurgy, and open-cast mining. The district municipality comprises seven local municipalities, specifically Dipaleseng, Govan Mbeki, Lekwa, Msukukaligwa, and DR Pixley ka Seme, all of which are situated within the Highveld priority area. The Chief Albert Luthuli and Mkhondo municipalities are not encompassed under the Highveld priority area. Figure 1 illustrates the geographical distribution of seven local district municipalities, with Gert Sibande being visually distinguished by the use of light-yellow highlighting.

### 2.2. Study Population, Sample Size Estimation, and Sampling Procedure

The participants in this study consisted of preschool-aged children, ranging from one to eight years old, who resided in and attended preschools located in either rural or urban areas within the Mpumalanga province, specifically in the Gert Sibande district municipality. Based on the data from the 2019 Gert Sibande database, the number of children enrolled in preschool was recorded as 13,485 (see Appendix A) [13]. The overall sample size required for this study was determined to be 3900, assuming a response rate of 70%. A study power of 80% was used for the investigation, with a significance level of 5%. The sample size was determined using the sample size calculator in Microsoft Excel (see Appendix A).

A probability sample design was employed to achieve equitable representation of all preschool children throughout seven local municipalities (see Appendix A). Preschools were identified in the northern, southern, eastern, and western regions of each of the seven municipalities within the Gert Sibande district (Figure 1). A representative sample of preschools was chosen from each of the four areas within each municipality. Preschool children were selected randomly from a class roster obtained from each designated preschool. Selected preschoolers were then given participant information leaflets inviting their parents to be part of the study. Parents, who consented to let their children participate, were then given an informed consent form and a questionnaire to complete (during parent meetings or when they pick up or drop off the children) and return to the preschool. All necessary COVID-19 protocols were implemented.

### 2.3. Study Tools

Data were collected using the modified International Study of Asthma and Allergies in Childhood (ISAAC) questionnaire. The questionnaire was divided into two sections: namely demographic data and health outcomes. The questionnaire was in English, which is the common language in a region with multiple local languages. In order to evaluate the data collection procedure and the quality of the survey questions in light of COVID-19 limitations, a pre-test of the instrument was conducted with environmental health practitioners. These professionals were chosen owing to their regular interaction with parents and carers, which provides them with insights into the educational background of these individuals.

### 2.4. Health Outcomes of the Study

The following central questions about asthma symptoms were used in order to evaluate health outcomes: (1) Has your child ever experienced chest wheezing or whistling in the past? (Wheeze Ever) (2) Has he or she had chest wheezing or whistling in the previous 12 months? (Current Wheeze) (3) How many wheezing episodes did your child have in the last 12 months? (4) How frequently, on average, during the previous 12 months was your child’s sleep interrupted by wheezing? (5) Has your child’s wheezing ever been sufficiently severe to prevent them from speaking more than a few phrases at any time between breathing in the last 12 months? (6) Did the child ever suffer from asthma? (Asthma Ever) (7) Did a physician or nurse diagnose the asthma? (8) Has your child’s chest ever made a wheezy noise while playing or right after? (9) Besides from a cough brought on by a cold or chest illness, has your child experienced a cough that is dry at night in the last 12 months?

The classification of Current Severe Wheeze was determined if parents provided affirmative responses to every one of the subsequent questions, noting that (1) the children have severe wheezing, with a frequency of 4–12 bouts or over 12 episodes throughout the preceding 12-month period; (2) the children experience disrupted sleep as a result of wheezing at least once a week or more; (3) the children had experienced a wheezing episode within the last 12 months, resulting in a reduction in their ability to speak to just a few phrases at a time due to intermittent breaths; and (4) the children experienced wheezing symptoms during or following physical activity throughout the preceding 12-month period.

### 2.5. Environmental Tobacco Smoke Exposure

Parents and caregivers were requested to provide responses pertaining to risk factors associated with wheezing, a symptom commonly observed in individuals with asthma. The questions encompassed the following: Does the male parent engage in smoking behaviour? (yes/no). Is the female parent engaged in the act of smoking? (yes/no).

The present study examines the extent of children’s exposure to smoking inside their household during the past 30 days, categorized into several frequency levels: never, 1–6 days, 7–10 days, 16–20 days, and more than 20 days. The present study examines the prevalence of children’s exposure to smoking inside the school environment over a period of 30 days.

The duration of the observed time intervals ranges from never through 1–6 days, 7–10 days, 16–20 days, and more than 20 days. The present study examines the extent to which children have been exposed to smoking in cars or other modes of transportation within the preceding 30 days. The duration of the event might vary, ranging from less than a week to over three weeks. The present study examines the frequency of children being subjected to smoking within the past 30 days in a restaurant setting. Parents/caregivers were asked to indicate the number of days in which children were exposed to smoking, with response options ranging from never to more than 20 days. In addition, they were asked the following question: what is the number of individuals residing in the same household as your child that engages in smoking?

### 2.6. Confounders

Parents and caregivers were asked to answer a series of questions about the following topics: What is the gender of the child (male/female)? What is the location of the child (rural/urban)? How long has the child lived in the area (6 to 12 months/1 to 2 years/3 years or longer)? Was the child born in the area (hospital/clinic/home/does not apply)? What kind of residence does the child live in (brick/mud/corrugated iron/mixture/other)? In the last 12 months, has the child used analgesics/antibiotics (never/at least once a year/at least once per month)? What type of fuel is utilized for cooking and heating (electricity/gas/paraffin/coal/wood/other)? How does the child get to and from school (walks/taxi/bus/motor vehicle/combination/other)? How frequently do trucks, buses, and taxis pass through your neighbourhood (never/rarely/frequently throughout the day/almost the entire day)? Other questions focused on pet ownership, education for parents/caregivers, job occupation of parents/caregivers, and family health history.

### 2.7. Data Processing and Analysis

The data were captured using EpiData version 3.1 [14], for the purpose of ensuring quality, and subsequently analysed using STATA 17. Descriptive statistics were computed, utilizing means and standard deviations for continuous data, and frequencies expressed as percentages for categorical data. Observations that were labelled as “not recorded” were designated as missing. Consequently, there were variations in the sample sizes utilized to address each respective question.

In this study, we assessed the association between demographic factors, including gender, age, location, and family history, with four outcome variables: Wheeze Ever, Current Wheeze, Current Severe Wheeze, and Asthma Ever. Statistical comparisons were conducted using the chi-square test for independent samples. The researchers employed multiple logistic regression to account for any confounding variables, assessing the strength of the relationship using the odds ratio (OR) and 95% confidence intervals (CIs). For outcomes with two categories, binary multiple logistic regression was utilized.

## 3. Results

### Description of Study Participants

We identified 3900 preschoolers and invited their parents, using participant information leaflets, to be part of the study. Three-thousand one-hundred and forty-five parents permitted their children and consented to participate, which was a participation rate of 80.6%. The preschoolers were, on average, 4.05 (SD = 1.22) years old. Most preschoolers were within the age range of 3 to 5 years, which fell within the 50th percentile. There were 1605 (51%) boys and 1540 (48.9%) girls. Most preschoolers (75%) resided in rural areas, while 773 (24.5%) lived in urban areas. Moreover, a significant majority of preschool-aged children (87%) were born in hospitals located in suburbs or township areas. Additionally, a substantial proportion of these children (80%) resided in these suburban or township regions for a duration of three years or more.

Table 1 provides a concise overview of the basic characteristics exhibited by children while Table 2 presents the environmental tobacco smoke exposure sources and health outcomes of the study participants. Study findings revealed that 23.56% of preschool children had male parents who engaged in smoking, whereas just 3.10% of preschool children had female parents who engaged in smoking. According to Table 2, a total of 28.86% of preschoolers resided in households where one or more individuals engaged in smoking activities within the same living space.

The study found that the overall prevalence of Wheeze Ever among the preschoolers was 15.14%, with a greater prevalence observed among urban preschoolers compared to their rural counterparts (20.7% vs. 13.3%, *p* < 0.001). Moreover, the total prevalence of Asthma Ever was 2.34%. The prevalence was also greater in urban preschoolers compared to rural preschoolers (3.9% vs. 1.8%, *p* < 0.001). The prevalence of Current Wheeze was found to be higher than that of Current Severe Wheeze and Asthma Ever, as indicated in Table 2.

The prevalence rates of Wheeze Ever, Current Wheeze, and Asthma Ever among urban preschoolers residing in households with one or more individuals who engage in smoking were found to be 23.11%, 14.14%, and 5.97%, respectively. In comparison, their rural counterparts exhibited prevalence rates of 17.15%, 12.93%, and 2.32% for the same respiratory conditions.

Furthermore, urban preschool children exposed to smoking at restaurants in the past 30 days had a 37.50% prevalence rate of Current Wheeze, while their rural counterparts had a prevalence of 11.32% for the same exposure days. Contrary to the above, it was observed that rural preschool children who had a female parent or caregiver who smoked exhibited a significantly higher prevalence of Current Wheeze, with a rate of 26.31%. The data presented in Table 3 illustrate the relevant information pertaining to the topic at hand.

The prevalence of Wheeze Ever and Current Wheeze was higher in children who had a pet in the house as compared to those without (Table 4 and Table 5). Moreover, the prevalence of Wheeze Ever in both rural and urban areas combined exhibited a greater incidence among boys (16.73%) compared to girls (13.49%). The data presented in Table 4 and Table 6 indicate that there is a larger prevalence of Current Severe Wheeze among boys (1.75%) compared to girls (0.78%) in both rural and urban areas.

Table 3 shows the multiple logistic regression analysis of risk factors for Wheeze Ever, Current Wheeze, Current Severe Wheeze and Asthma Ever for rural and urban areas, with their respective odds ratios. Both urban and rural area children who lived with one or more people who smoked in the same house (WE: OR 1.44, 95% CI 1.11–1.86) (CW: OR 2.09, 95% CI 1.38–3.16) and (AE: OR 2.49, 95% CI 1.12–5.54) were found to have an increased likelihood for having Wheeze Ever, Current Wheeze and Asthma Ever as compared to those who lived with non-smokers. Moreover, those children exposed to smoking at the restaurant for one or more days in the past 30 days (CW: OR 2.27, 95% CI 1.17–4.38) were more likely to present with Current Wheeze as compared to those who had never been exposed to smoking at the restaurant.

In the context of combined rural and urban areas, Wheeze Ever and Current Wheeze shared similar ETS risk factors. The occurrence of symptoms was shown to be significantly higher in cases when a female parent or caregiver engaged in smoking behaviour. The crude odds ratios (ORs) for Wheeze Ever and Current Wheeze were 3.11 (95% CI 2.00–4.83), and 3.12 (95% CI 1.90–5.12), respectively. In addition, there was a notable relationship between the number of smoking individuals residing in the same household as preschoolers and the likelihood of developing Wheeze Ever, Current Wheeze, and Current Severe Wheeze. The adjusted odds ratios for these associations were 1.37 (1.08–1.74), 2.09 (1.38–3.16), and 2.46 (1.25–4.85), respectively, as indicated in Table 4, Table 5 and Table 6.

Some of the confounders that showed significant associations with Wheeze Ever in both rural and urban areas were children preschoolers using analgesic/antibiotics in the past 12 months at least once a year (adjusted OR 2.29, 95% CI 1.41–3.71) and preschoolers using a motorcar as their mode of transportation to school (adjusted OR 1.74, 95% CI 1.27–2.38) (refer to Table 4). The male gender was shown to be associated with a higher probability of experiencing both Wheeze Ever (OR 1.35, 95% CI 1.08–1.70) and Current Severe Wheeze (OR 2.30, 95% CI 1.09–4.84) according to the data presented in Table 4 and Table 6. Having a female parent or caregiver who worked in the government sector was shown to be associated with an elevated probability of experiencing Current Wheeze, as indicated by an odds ratio of 1.64 (95% CI 1.06–2.54), as presented in Table 5. The presence of a dog in the household during a period of 12 months has been found to be associated with an increased probability of experiencing Current Wheeze (OR 1.79, 95% CI 1.28–2.51), according to the crude odds ratios reported in Table 5.

## 4. Discussion

This study aimed to assess the prevalence of wheezing and its association with environmental tobacco smoke exposure, a common symptom of asthma, among preschool children residing in rural and urban areas of Mpumalanga province, South Africa. The reported prevalence of wheeze in Mpumalanga is similar to that observed in previous research. Based on the findings of the ISAAC Phase Three study, it was determined that the worldwide prevalence of current wheeze among school-aged children was 11.5%. This prevalence showed significant regional variation, varying from 6.8% in the Indian subcontinent to 21% in Oceania [15]. Furthermore, the prevalence of Current Wheeze (10%) and lifetime asthma (3.4%) in Africa exhibited a comparable pattern to the outcomes observed in our study [15]. Additionally, the findings of Huq et al. [16] who assessed allergy symptoms and diagnosis among children aged three years and five months from the Venda Health Examination of Mothers, Babies and their Environment (VHEMBE), Limpopo province in South Africa, support our study results. The prevalence of wheezing (4.7%) observed in their study aligns with the findings of our study.

The potential influence of various factors on the prevalence of asthma symptoms within a given region can be attributed to several key variables, including the age range of children considered in the study, the prevailing climate conditions, the specific timing of the study, the size of the sampled population, the design of the study itself, and the presence or absence of certain risk factors. Research studies that specifically examine children within similar age groups, as well as children residing in a particular place for a duration beyond six months, have found comparable rates of asthma symptom prevalence. Based on the aforementioned findings, it is evident that the management of asthma symptoms poses a persistent problem. Consequently, it may be necessary to formulate and execute strategies aimed at mitigating these symptoms within this specific demographic promptly.

This study found that there was a higher prevalence of Wheeze Ever and Asthma Ever among preschoolers living in urban areas compared to those residing in rural areas. Consistent with the results of our study, Levin et al. [17], Feng et al. [18], Rodriguez et al. [19], and Kutzora et al. [11] conducted research in South Africa, in China, a systematic review, and in Germany, respectively, which also indicated a greater prevalence of asthma and or symptoms among children residing in urban regions compared to those dwelling in rural areas. The present study found that children residing in the Mpumalanga Highveld region were predominantly impacted by Wheeze Ever, a common symptom of asthma, and also had a history of Asthma Ever, particularly if they attended an urban preschool. The results of our study align with the majority of the existing literature, which consistently demonstrates that residing in rural areas or on farms, being exposed to livestock, and the hygiene hypothesis confer protective advantages against the development of asthma and or symptoms in childhood compared to children residing in urban areas [11,20]. Additionally, our research findings provide support for the notion that children residing in urban areas are more prone to increased exposure and heightened sensitivity to several risk factors associated with asthma and or symptoms [21].

The study outcome indicated above may have been influenced by specific environmental factors. The regions of Mpumalanga Highveld exhibit a notable deterioration in air quality, with heightened levels of pollutants stemming from both industrial and non-industrial origins. The district encompasses a variety of sectors, namely power generating, petrochemical, primary metallurgy, and open-cast mining. Urban environments possess a multitude of modifiable exposures that can impact the prevalence and morbidity of asthma symptoms. In the aggregate of both rural and urban areas, boys had a greater propensity for experiencing Wheeze Ever at any point and Current Severe Wheeze in comparison to their girl counterparts.

This observation aligns with previous research indicating that boys consistently have a higher incidence of wheezing and/or asthma symptoms relative to girls [4,22].

Risk factors and confounders associated with wheeze, a symptom commonly observed in individuals with asthma, were identified in our study. The risk factors with the highest potential for modification encompassed a female parent who engages in smoking, a male parent who engages in smoking, the number of individuals residing in the same household as a child and who engage in smoking, exposure to smoking within the home (within the previous 30 days), exposure to smoking within a motor vehicle or transportation (within the previous 30 days), exposure to smoking within a restaurant (within the previous 30 days), the mode of transportation utilized to commute to school, and ownership of a pet.

This study found that preschoolers were more likely to experience Wheeze Ever and Current Wheeze in their lives, if they had a female parent or caregiver who smoked, and if they lived in the same household as one or more people who smoked. Those who were exposed to smoking in cars and restaurants in the past 30 days were more likely to present with Current Wheeze. The results of our study are consistent with the existing literature, which indicates that children are primarily exposed to environmental tobacco smoke (ETS) through smoking by adults in environments where children reside and engage in recreational activities. This exposure significantly increases their susceptibility to developing asthma and or symptoms [23,24,25].

According to studies conducted by Farzan et al. [26] and Shahunja et al. [27], there exists a significant relationship between the prevalence of asthma symptoms in children and their exposure to household tobacco smoke. Moreover, Wang et al. [28] conducted a study that revealed a significant relationship between the presence of wheezing symptoms in children and their exposure to second-hand smoke. In addition, Tabuchi et al. [29] and Harju et al. [30] also reported that children who had two smoking parents were more likely to have asthma symptoms and had a greater chance of asthma attacks relative to children with non-smoking parents. Simic et al. [31] provided additional support for the aforementioned results since they demonstrated that there are significant differences in the episodes of asthma symptoms in children depending on the area or room within which parents smoked the cigarettes.

Although the presence of second-hand smoke has been identified as a significant indicator of asthma symptoms, there remains a lack of consensus regarding the specific threshold at which exposure to smoking becomes detrimental. It is thus highly advisable to completely refrain from exposure to second-hand smoke and to ensure that household members who smoke confine their smoking activities to isolated areas that are inaccessible to these children. Parents should additionally take into consideration the implementation of a prohibition on smoking within the confines of their residence or its immediate vicinity.

Additionally, our study revealed that preschool-aged children who have been subjected to ETS in cars or transport without a complete physical barrier within the last 30 days were shown to have a higher likelihood of experiencing Current Wheeze. In addition, our research revealed an association between the utilization of motor vehicles for transportation to school among preschool-aged children and an increased likelihood of experiencing Wheeze Ever.

The anticipated outcomes of this study are in line with expectations, as the act of parents or caregivers smoking in the car during the transportation of children to school has been found to elevate exposure to ETS and therefore raise the probability of experiencing symptoms associated with asthma. The existing body of literature on the exposure of children to ETS and its impact on the development of respiratory and asthma symptoms provides substantial data that aligns with the findings of the aforementioned study [24,32]. Additionally, the use of motor vehicles may contribute to an increased likelihood of experiencing symptoms associated with asthma. Gasana and colleagues [33] conducted a study that corroborated the aforementioned findings, as they concluded that children who are exposed to elevated amounts of air pollution from motor vehicles are more likely to exhibit symptoms of childhood wheezing. Lau et al. [34] reported that traffic-related exposure tends to be closely related to asthma and or wheezing in children. Moreover, Suhaimi and colleagues [35] found that children in elevated-traffic areas were four times more likely to present with wheezing as compared to children in low-traffic areas. It is advisable to prioritize the avoidance of ETS exposure as a crucial factor in mitigating the onset and facilitating the control of asthma and related symptoms.

## 5. Strength and Limitation of the Study

Firstly, the ISAAC questionnaire is a valid tool for data collection for this investigated population group and has been utilized worldwide in studies investigating asthma symptoms. Secondly, this study had a great participation rate with over 3000 children, which is a requirement of ISAAC centres, thus increasing the study’s statistical power. Finally, the implementation of a standardized and validated tool facilitates the ability to compare study findings with those of other studies conducted at various levels, including local, regional, and international contexts.

The study outcomes may deviate slightly from the actual prevalence of investigated symptoms due to the presence of missing data. Future research endeavours should prioritize the meticulous completion of questionnaires, aiming to minimize the occurrence of missing data to a significant extent. The study gathered data from the past year by using a parental-completed questionnaire. It was anticipated that these parents, who primarily reside with the children, would be able to accurately recall the information requested. The one-year timeframe was considered sufficient for recollection, without posing significant obstacles. The study was conducted during the COVID-19 period which had an impact on the implementation of the study methodology due to limited accessibility to the study areas and the availability of preschoolers to participate in the study.

## 6. Conclusions

The study found that in Mpumalanga, preschoolers living in urban areas had a higher prevalence of Wheeze Ever, Current Wheeze, Current Severe Wheeze, and Asthma Ever relative to rural preschoolers. The presence of ETS exposure among preschool-aged children in various settings, including at restaurants and during transportation, increased the probability of experiencing wheezing. The implementation of smoking limits and prohibition is crucial in areas that are frequented or utilized by children. Hence, it is imperative for healthcare providers to actively champion the rights of individuals who do not smoke within society, while also endorsing legislative measures aimed at curtailing tobacco smoke exposure.

## Figures and Tables

**Figure 1 ijerph-21-00469-f001:**
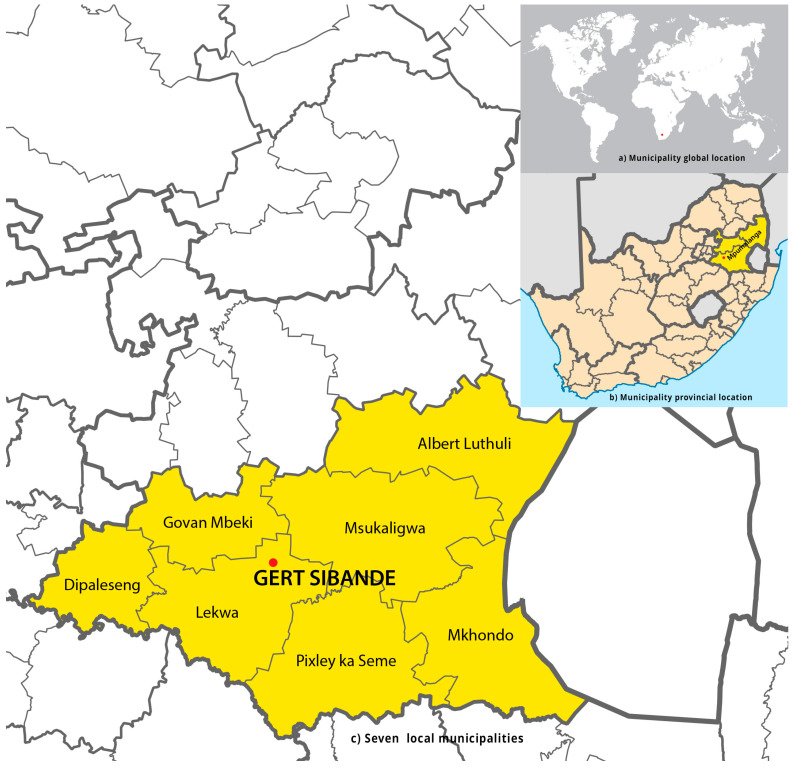
Geographical distribution of preschools within the Gert Sibande district municipality. (**a**) provides a visual representation of the global location of the preschools within the Gert Sibande municipality. (**b**) presents the spatial distribution of the Gert Sibande municipality in the province of Mpumalanga within the broader context of the nine provinces of South Africa. (**c**) is an illustration depicting the inclusion of all seven local municipalities within the Gert Sibande district, wherein preschools were identified, highlighted in a light-yellow colour. The red dot on the map indicates the municipality’s location.

**Table 1 ijerph-21-00469-t001:** The demographic characteristics of the participants in the study (*n* = 3145).

Variables	*N*	Percentage (%)
1. Gender of the child		
Female	1540	48.97
Male	1605	51.03
2. Age group of children		
<3 years	414	13.16
3–5 years	1779	56.57
≥5 years	952	30.27
3. Child location		
Rural	2372	75.42
Urban	773	24.58
4. Time lived in suburb/township		
Less than 6 months	107	3.40 (3.49)
6 to 12 months	99	3.14 (3.23)
1 to 2 years	408	12.9 (13.32)
3 years or longer	2450	77.90 (79.96)
Missing	81	2.57
5. Type of house the child lives in ^a^		
Brick	2547	80.98 (82.37)
Mud	116	3.68 (3.75)
Corrugated iron	255	8.10 (8.25)
Combination	93	2.95 (3.01)
Other	81	2.57 (2.62)
Missing	53	1.68
6. Fuel used for cooking in the house ^b^		
Electricity	2476	78.72 (78.78)
Gas	100	3.17 (3.18)
Paraffin	19	0.60 (0.60)
Coal	254	8.07 (8.08)
Wood	292	9.28 (9.29)
Other	2	0.06 (0.06)
Missing	2	0.06
7. Fuel used for heating in the house ^c^		
Electricity	2008	63.84 (64.44)
Gas	198	6.29 (6.35)
Paraffin	27	0.85 (0.87)
Coal	484	15.38 (15.53)
Wood	380	12.08 (12.20)
Other	19	0.60 (0.61)
Missing	29	0.92
8. Child used analgesic/antibiotic in the past 12 months		
Never	445	14.14 (14.59)
At least once a year	921	29.28 (30.21)
At least once per month	1683	53.51 (55.20)
Missing	96	3.05
9. How does the child get to school ^d^		
Walk	1666	52.97 (53.41)
Taxi/bus	848	26.96 (27.19)
Motor car	525	16.69 (16.83)
Combination	57	1.81 (1.83)
Other	23	0.73 (0.74)
Missing	26	0.82
10. How often has your child been absent from school (past 6 months)		
Never or occasionally	1854	58.95 (61.51)
Once or twice per week	904	42.14 (29.99)
Three or more times a week	256	8.13 (8.49)
Missing	131	4.1
11. Child ever breastfed		
No	974	30.96 (31.43)
Yes	2125	67.46 (68.57)
Missing	46	1.46
12. Truck traffic pass through the street on weekdays		
Never	518	16.47 (16.99)
Seldom	735	23.37 (24.11)
Frequent through the day	713	22.67(23.38)
Almost all day	1083	34.43 (35.52)
Missing	96	3
13. Female parent: highest level of school completed ^e^		
Primary	246	7.82 (8.18)
Secondary	1884	59.90 (62.67)
University	454	14.43 (15.10)
Other	422	13.41 (14.04)
Missing	139	4.41
14. Female parent job industry		
Government sector	351	11.16 (11.76)
Private sector	580	18.44 (19.44)
Self-employed	308	9.79 (10.32)
Not employed	1745	55.48 (58.48)
Missing	161	5.11
15. Female parent ever asthma		
No	2487	79.09 (96.92)
Yes	79	2.51 (3.08)
Missing	579	18.41
16. Cat inside the house		
No	2885	91.17 (92.82)
Yes	223	7.09 (7.18)
Missing	37	1.17
17. Dog inside the house		
No	2780	88.39 (89.36)
Yes	331	10.52 (10.64)
Missing	34	1.08

( ) *Missing data were excluded from the data analysis.*
^a^ Combination includes brick and corrugated iron: other includes wood. ^b^ Other includes generator. ^c^ Other includes solar energy/electricity. ^d^ Combination includes motorcar and taxi/bus, and other includes animal cart. ^e^ Other refers to all types of informal education.

**Table 2 ijerph-21-00469-t002:** Environmental tobacco smoke exposure sources and health outcomes of the study participants (*n* = 3145).

Variables	*N*	Percentages (%)
1. Female parent smokes		
Yes	94	2.98 (3.10)
No	2934	93.29 (96.90)
Missing	117	3.72
2. Male parent smokes		
Yes	451	14.34 (23.56)
No	1463	46.51 (76.44)
Missing	1231	39.14
3. How many people living in the same house as your child smoke?		
Zero	2051	65.21 (71.14)
One or more	832	26.45 (28.86)
Missing	262	8.33
4. Child exposure to smoking at home (past 30 days)		
Never	1947	61.90 (83.10)
One or more days *	396	12.59 (16.90)
Missing	802	25.5
5. Child exposure to smoking at school (past 30 days)		
Never	1444	45.91 (97.30)
One or more days *	40	1.27 (2.70)
Missing	1661	52.81
6. Child exposure to smoking in car/transport (past 30 days)		
Never	1390	44.19 (95.21)
One or more days *	70	2.25 (4.79)
Missing	1685	53.57
7. Child exposure to smoking at the restaurant (past 30 days)		
Never	1387	44.10 (94.61)
One or more days *	79	2.51 (5.39)
Missing	1678	53.35
8. Wheeze Ever		
Yes	467	14.8 (15.14)
No	2617	83.2 (84.86)
Missing	61	1.9
9. Current Wheeze		
Yes	292	9.2 (9.45)
No	2799	88.9 (90.55)
Missing	54	1.7
10. Current Severe Wheeze		
Yes	40	1.27 (1.28)
No	3076	97.8 (98.72)
Missing	29	0.92
11. Asthma Ever		
Yes	66	2.09 (2.34)
No	2810	89.34 (97.65)
Missing	269	8.55
12. Wheeze attack in the past 12 months		
Yes	274	8.71 (8.87)
No	2815	89.50 (91.13)
Missing	56	1.78
13. Sleep disturbed due to wheeze in the past 12 months		
Yes	199	6.32 (6.45)
No	2888	91.82 (93.55)
Missing	58	1.82
14. Wheeze severe enough to limit speech in the past 12 months		
Yes	59	1.87 (1.92)
No	3016	95.89 (98.08)
Missing	75	2.38
15. Asthma diagnosed by a medical doctor or nurse		
Yes	54	1.71 (1.81)
No	2929	93.13 (98.19)
Missing	162	5.15
16. Chest ever sounded wheezy during/after playing		
Yes	232	7.37 (8.82)
No	2398	76.24 (91.18)
Missing	515	16.37
17. Dry cough at night apart from cough associated with cold or chest infection		
Yes	650	20.66 (22.58)
No	2229	70.87 (77.42)
Missing	266	8.45

( ) Missing data were excluded from the data analysis. * One or more days include the following day range: 1–6 days; 7–10 days; 16–20 days; more than 20 days.

**Table 3 ijerph-21-00469-t003:** Participants’ prevalence of Wheeze Ever, Current Wheeze, Current Severe Wheeze and Asthma Ever for rural and urban areas with their respective odds ratios.

	Rural						Urban					
Variable	Total ^a^	Prevalence (%)	Crude OR ^b^		Adjusted OR ^b^		Total ^a^	Prevalence (%)	Crude OR ^b^		Adjusted OR ^b^	
			(95% CI)	*p*	(95% CI)	*p*			(95% CI)	*p*	(95% CI)	*p*
Wheeze Ever ^c^												
How often have you given your child medication (past 12 months)?												
Never	351	4.55	1		1		84	8.33	1		1	
At least once a year	611	12.43	2.78 (1.75–4.41)	**0.000**	2.31 (1.36–3.92)	**0.002**	298	18.45	2.78 (1.75–4.41)	**0.000**	2.31 (1.36–3.92)	**0.002**
At least once per month	1300	16	3.98 (2.56–6.17)	**0.000**	3.03 (1.83–5.02)	**0.000**	350	36.18	3.98 (2.56–6.17)	**0.000**	3.03 (1.83–5.02)	**0.000**
Do people living in the same house as your child smoke?												
Zero	1500	11.4	1		1		511	20.15	1		1	
One or more	618	17.15	1.46 (1.17–1.82)	**0.001**	1.43 (1.11–1.85)	**0.006**	199	23.11	1.46 (1.17–1.82)	**0.001**	1.43 (1.11–1.85)	**0.006**
Cat inside the house (past 12 months)												
No	2143	13.06	1		1		687	19.65	1		1	
Yes	156	16.02	1.54 (1.09–2.17)	**0.012**	1.34 (0.87–2.08)	0.182	64	34.37	1.54 (1.09–2.17)	**0.012**	1.34 (0.87–2.08)	0.182
Female parent level of school completion												
Primary	222	18.01	1		1		19	23.31	1		1	
Secondary	1523	11.81	0.59 (0.41–0.84)	**0.004**	0.76 (0.46–1.25)	0.290	323	16.09	0.59 (0.41–0.84)	**0.004**	0.76 (0.46–1.25)	0.290
University	212	16.50	1.03 (0.68–1.55)	0.871	1.43 (0.83–2.43)	0.192	237	28.27	1.03 (0.68–1.55)	0.871	1.43 (0.83–2.43)	0.192
Other	286	15.73	0.82 (0.54–1.24)	0.354	0.99 (0.53–1.63)	0.820	127	21.25	0.82 (0.54–1.24)	0.354	0.99 (0.53–1.63)	0.820
Female parent ever had asthma												
No	1852	12.41	1		1		599	18.19	1		1	
Yes	43	25.58	3.65 (2.28–5.85)	**0.000**	2.95 (1.79–4.88)	**0.000**	36	55.55	3.65 (2.28–5.85)	**0.000**	2.95 (1.79–4.88)	**0.000**
Current Wheeze ^d^												
Female parent ever had asthma												
No	1853	8.04	1		1		600	9.33	1		1	
Yes	43	23.25	3.82 (2.26–6.45)	**0.000**	5.14(2.55–10.34)	**0.000**	36	30.55	3.82 (2.26–6.45)	**0.000**	5.14 (2.55–10.34)	**0.000**
Child used analgesic/antibiotic in the past 12 months												
Never	354	1.97	1		1		86	1.16	1		1	
At least once a year	609	7.38	3.85 (1.82–8.12)	**0.000**	3.41 (1.17–9.95)	**0.024**	297	6.06	3.85 (1.82–8.12)	**0.000**	3.41 (1.17–9.95)	**0.024**
At least once per month	1307	11.40	7.94 (3.89–16.22)	**0.000**	4.74 (1.68–13.40)	**0.003**	349	18.33	7.94 (3.89–16.22	**0.000**	4.74 (1.68–13.40)	**0.003**
Female parent job industry												
Government sector	183	11.47	1		1		164	14.63	1		1	
Private sector	371	7.27	0.62 (0.40–0.96)	**0.034**	0.78 (0.41–1.48)	0.462	196	10.20	0.62 (0.40–0.96)	**0.034**	0.78 (0.41–1.48)	0.462
Self-employed	199	9.04	0.76 (0.46–1.25)	0.292	0.86 (0.42–1.73)	0.678	102	11.76	0.76 (0.46–1.25)	0.292	0.86 (0.42–1.73)	0.678
Not employed	1476	9.01	0.73 (0.50–1.06)	0.103	0.58 (0.32–1.03)	0.066	245	10.20	0.73 (0.50–1.06)	0.103	0.58 (0.32–1.03)	0.066
Dog inside the house (past 12 months)												
No	2103	8.13	1		1		639	10.79	1		1	
Yes	205	15.60	1.74 (1.24–2.44)	**0.001**	1.29 (0.71–2.33)	0.393	115	13.04	1.74 (1.24–2.44)	**0.001**	1.29 (0.71–2.33)	0.393
Female parent smoke												
No	2186	8.26	1		1		698	11.31	1		1	
Yes	76	26.31	3.19 (1.94–5.24)	**0.000**	0.93 (1.31–2.75)	0.901	17	11.76	3.19 (1.94–5.24)	**0.000**	0.93 (1.31–2.75)	0.901
Do people living in the same house as your child smoke?												
Zero	1497	7.34	1		1		511	9.39	1		1	
One or more	626	12.93	1.77 (1.36–2.30)	**0.000**	2.09 (1.36–3.110)	**0.001**	198	14.14	1.77 (1.36–2.30)	**0.000**	2.06 (1.36–3.10)	**0.001**
Child exposure to smoking at the restaurant (past 30 days)												
Never	984	8.23	1		1		381	11.81	1		1	
One or more days	53	11.32	2.35 (1.29–4.26)	**0.005**	2.28 (1.18–4.41)	**0.014**	24	37.5	2.35 (1.29–4.26)	**0.005**	2.28 (1.18–4.41)	**0.014**
Current Severe Wheeze ^e^												
Dry cough at night apart from cough associated with cold or chest infection												
No	1664	0.30	1		1		564	0.17	1		1	
Yes	492	4.06	16.75 (6.90–40.64)	**0.000**	13.44 (4.51–40.0)	**0.000**	156	5.12	16.75 (6.90–40.64)	**0.000**	13.44 (4.51–40.0)	0.000
Dog inside the house (past 12 months)												
No	2119	1.13	1		1		642	0.93	1		1	
Yes	209	2.39	2.60 (1.21–5.55)	**0.014**	1.43 (0.46–4.45)	0.529	116	3.44	2.60 (1.21–5.55)	**0.014**	1.43 (0.46–4.45)	0.529
Male parent smoke												
No	1017	0.98	1		1		431	0.92	1		1	
Yes	321	2.49	2.57 (1.15–5.70)	**0.020**	1.66 (0.59–4.66)	0.332	128	2.34	2.57 (1.15–5.70)	**0.020**	1.66 (0.59–4.66)	0.332
Do people living in the same house as your child smoke?												
Zero	1515	0.99	1		1		513	0.97	1		1	
One or more	628	2.22	2.23 (1.17–4.24)	**0.014**	1.62(0.56–4.65)	0.364	199	2.01	2.23 (1.17–4.24)	**0.014**	1.62 (0.56–4.65)	0.364
Asthma Ever ^f^												
Do people living in the same house as your child smoke?												
Zero	1364	1.31	1		1		479	3.54	1		1	
One or more days	558	2.32	175 (1.03–2.98)	**0.036**	1.79 (0.99–3.26)	**0.054**	184	5.97	175 (1.03–2.98)	**0.036**	1.79 (0.99–3.26)	**0.054**
Child used analgesic/antibiotic in the past 12 months												
Never	314	0.31	1		1		76	1.31	1		1	
At least once a year	557	1.97	3.14 (0.71–13.85)	0.130	3.52 (0.44–27.90)	0.233	277	1.44	3.14 (0.71–13.85)	0.130	3.52 (0.44–27.90)	0.233
At least once per month	1176	0.22	6.14 (1.48–25.42)	**0.012**	7.70 (1.04–56.7)	**0.045**	328	6.40	6.14 (1.48–25.42)	**0.012**	7.70 (1.04–56.7)	**0.045**
Cat inside the house (past 12 months)												
No	1937	1.70	1		1		636	3.30	1		1	
Yes	145	3.44	2.7 (1.44–5.25)	**0.002**	2.62 (1.20–5.68)	**0.015**	62	11.29	2.7 (1.44–5.25)	**0.002**	2.62 (1.20–5.68)	**0.015**
Female parent ever had asthma												
No	1730	1.56	1		1		560	3.21	1		1	
Yes	39	15.3	7.63 (3.72–15.66)	**0.000**	4.37 (1.90–10.04)	**0.000**	35	14.28	7.63 (3.72–15.66)	**0.000**	4.37 (1.90–10.04)	**0.000**

^a^ The totals for each risk factor are different due to differences in missing values. Missing data were excluded from the data analysis. ^b^ The values that are statistically significant for the crude OR and less than 0.05 for the adjusted OR are highlighted in bold (only risk factors and confounders that showed association with health outcome were included in crude OR and adjusted OR). ^c^ The model was adjusted for all significant variables from crude OR. ^d^ The model was adjusted for all significant variables from crude OR. ^e^ The model was adjusted for all significant variables from crude OR. ^f^ The model was adjusted for all significant variables from crude OR.

**Table 4 ijerph-21-00469-t004:** Participants (combined rural and urban areas) prevalence of Wheeze Ever with their respective odds ratios.

Variable	Total ^a^	Prevalence (%)	Crude OR ^b^		Adjusted OR ^b,c^	
			(95% CI)	*p*	(95% CI)	*p*
Do people living in the same house as your child smoke?						
Zero	2011	13.60	1			
One or more	817	18.60	1.44 (1.16–1.40)	**0.001**	1.35 (1.07–1.71)	**0.011**
Female parent smoke						
No	2878	14.62	1		1	
Yes	92	34.78	3.11 (2.00–4.83)	**0.000**	2.68 (1.65–4.36)	**0.000**
Sex of child						
Female	1512	13.49	1		1	
Male	1572	16.73	1.28 (1.05–1.57)	**0.012**	1.34 (1.07–1.67)	**0.010**
Child used analgesic/antibiotic in the past 12 months						
Never	435	5.28	1		1	
At least once a year	909	14.41	3.01 (1.90–4.77)	**0.000**	2.43 (1.50–3.94)	**0.000**
At least once per month	1650	18.24	3.99 (2.57–6.19)	**0.000**	3.26 (2.06–5.15)	**0.000**
Cat in the house (past 12 months)						
No	2830	17.18	1		1	
Yes	220	21.36	1.58 (1.12–2.21)	**0.008**	1.37 (0.92–2.04)	0.111
Female parent level of school completion ^d^						
Secondary	1846	12.56	1			
University	449	22.71	2.04 (1.57–2.65)	**0.000**	1.86 (1.36–2.51)	**0.000**
Other	413	17.43	1.46 (1.10–1.96)	**0.009**	1.30 (0.94–1.79)	0.106
Primary	241	18.67	1.59 (1.12–2.27)	**0.009**	1.31 (0.85–2.03)	0.208
How does the child get to school? ^e^						
Walk	1633	13.16	1		1	
Taxi/bus	828	15.57	1.21 (0.96–1.54)	0.103	1.28 (0.98 –1.67)	0.066
Motor car	518	21.62	1.81 (1.41–2.34)	**0.000**	1.70 (1.25–2.31)	**0.001**
Combination	57	12.28	0.92 (0.41–2.06)	0.846	1.05 (0.45–2.42)	0.908
Other	23	4.34	0.29 (0.04–2.23)	0.240	0.98 (0.03–2.32)	0.252

^a^ The totals for individual risk factors differ owing to the absence of values. Missing data were excluded from the data analysis. ^b^ The statistically significant values for the crude OR and less than 0.05 for the adjusted OR are highlighted in bold (only risk factors and confounders that showed association with health outcome were included in crude OR and adjusted OR). ^c^ Model adjustments were made for all the variables in the table. ^d^ Other refers to all forms of informal education. ^e^ Combination includes motorcar and taxi/bus, for which other includes animal cart. 1: Unless declared in another manner, the referent category for individual risk factors is the lack of the risk factor.

**Table 5 ijerph-21-00469-t005:** Participants’ (combined rural and urban areas) prevalence of Current Wheeze with their respective odds ratios.

Variables	Total ^a^	Prevalence (%)	Crude OR ^b^		Adjusted OR ^b,c^	
			(95% CI)	*p*	(95% CI)	*p*
Female parent ever asthma						
No	2453	8.35	1		1	
Yes	79	26.58	3.97 (2.36–6.67)	**0.000**	5.59 (2.77–11.26)	**0.000**
Child used analgesic/antibiotic in the past 12 months						
Never	440	1.81	1		1	
At least once a year	906	6.95	4.03 (1.91–8.49)	**0.000**	3.41 (1.17–9.95)	**0.024**
At least once per month	1656	12.86	7.97 (3.90–16.27)	**0.000**	4.74 (1.68–13.40)	**0.003**
Female parent job industry						
Private sector	567	8.28	1		1	
Government sector	347	12.96	1.64 (1.06–2.54)	**0.024**	1.38 (0.73–2.60)	0.317
Self-employed	301	9.96	1.22 (0.75–1.98)	0.409	1.18 (0.73–2.32)	0.629
Not employed	1721	9.18	1.11 (0.79–1.57)	0.519	0.69 (0.41–1.17)	0.177
Dog in the house (past 12 months)						
No	2742	8.75	1		1	
Yes	320	14.68	1.79 (1.28–2.51)	**0.001**	1.27 (0.70–2.32)	0.419
Female parent smoke						
No	2884	9.01	1		1	
Yes	93	23.65	3.12 (1.90–5.12)	**0.000**	0.65 (0.20–2.15)	0.488
Do people living in the same house as your child smoke?						
Zero	2008	7.86	1		1	
One or more	823	13.12	1.78 (1.36–2.29)	**0.000**	2.09 (1.38–3.16)	**0.000**
Child exposure to smoking in the car (past 30 days)						
Never	1368	9.50	1		1	
One or more days	69	11.59	2.37 (1.31–4.30)	**0.004**	2.27 (1.17–4.38)	**0.014**

^a^ The totals for individual risk factors differ owing to the absence of values. Missing data were excluded from the data analysis. ^b^ The statistically significant values for the crude OR and less than 0.05 for the adjusted OR are highlighted in bold (only risk factors and confounders that showed association with health outcome were included in crude OR and adjusted OR). ^c^ Model adjustments were made for all the variables in the table. 1: Unless declared in another manner, the referent category for individual risk factors is the lack of the risk factor.

**Table 6 ijerph-21-00469-t006:** Participants’ (combined rural and urban areas) prevalence of Current Severe Wheeze with their respective odds ratios.

Variable	Total ^a^	Prevalence (%)	Crude OR ^b^		Adjusted OR ^b,c^	
			(95% CI)	*p*	(95% CI)	*p*
Do people living in the same house as your child smoke?						
Zero	2028	0.98	1		1	
One or more	827	2.17	2.23 (1.17–4.24)	**0.014**	2.46 (1.25–4.85)	**0.009**
Sex of child						
Female	1524	0.78	1		1	
Male	1592	1.75	2.25 (1.14–4.45)	**0.019**	2.30 (1.09–4.84)	**0.027**

^a^ The totals for individual risk factors differ owing to the absence of values. Missing data were excluded from the data analysis. ^b^ The statistically significant values for the crude OR and less than 0.05 for the adjusted OR are highlighted in bold (only risk factors and confounders that showed association with health outcome were included in crude OR and adjusted OR). ^c^ Model adjustments were made for all the variables in the table. 1: Unless declared in another manner, the referent category for individual risk factors is the lack of the risk factor.

## Data Availability

We did not receive ethics approval to share raw field data publicly. The data belong to the University of Pretoria (UP). The raw data analysed in the current study are available from UP on reasonable request.

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
