# Peer review of "Prevalence of Wheezing and Its Association with Environmental Tobacco Smoke Exposure among Rural and Urban Preschool Children in Mpumalanga Province, South Africa"

_ijerph, 2024, doi:10.3390/ijerph21040469_

Round 1

Reviewer 1 Report

Comments and Suggestions for Authors

The author explored the association between exposure to environmental tobacco smoke and the occurrence of wheezing among preschool children and compared the children’s prevalence of wheezing between rural and urban areas.

The title does not align the study's objectives.

Introduction:

1.     The study primarily emphasizes asthma rather than wheezing, despite wheezing being the outcome variable of interest.

2.     The new tobacco regulations were officially announced in September 2022, whereas this study was conducted between November 2020 and April 2021. Therefore, the relevance of these laws to the study should be to merely summarize the government’s actions to protect children from tobacco smoking (lines 61-79).

Method

1.     The author stated they use ‘probability sample design’ and this process includes some stages such as a municipality, regions, schools, and preschoolers. Several suitable sampling strategies could be done at each stage that can result in differences in the final sample size. The explanation for specific actions at each stage is necessary to be more clear.

2.     Using the method stated in the manuscript, there is a chance that children in different classes but from the same family were selected. This could introduce incorrect information by duplicated conditions that might be collected several times. With this concern, how do the authors manage to control for information bias?

3.     The instrument used for data collection was conducted with assistants from health care professionals, which is a good practice. However, the authors didn’t present how they controlled for the qualify of data collection in the main study. Based on the information in the manuscript, the questionnaires were just sent to parents and returned after being filled. Were there any methods to verify that the parents received the questionnaires? Did parents understand thoroughly the meaning intended by the questions?

4.     Regarding the health outcomes, ‘Asthma ever’ was determined based on the response to the question ‘Did the child ever suffer from asthma?’. However, the authors also asked parents ‘Did a physician or nurse diagnose the asthma?’. The diagnosis of asthma by healthcare providers was more reliable than the recognition of parents and parents were more likely aware of their child’s health from them. Can the authors explain their choice for using self-reported asthma as an outcome?

5.     The authors figured out the reason of choosing the Gert Sibande district in this study because its air quality and heightend levels of pollutants. However, this study aims to evaluate the association between exposure to environmental tobacco smoke and the occurrence of wheezing, a symptom commonly associated with asthma. I wonder if choosing this district might lead to interfere with the research results. (line 104-114)

6.     Regarding Figure 1, it would be preferable for the authors to create an original figure rather than using one sourced from the internet. Have the authors obtained permission from the website for its use?

7.     The labels a, b, c appear to be missing in Figure 4.1. Could the authors provide clarification on their whereabouts?

8.     It is unclear why the first figure in the study is labeled as Figure 4.1. An explanation for this numbering choice would be beneficial.

9.       The methodology for determining the overall sample size (lines 128-129) needs clarification. What criteria were used to arrive at this number?

10.  Incorporating a diagram to elucidate the sampling procedure would greatly enhance understanding.

11.  “The questionnaire was in English, which is the common language in a region with multiple local languages” à It raises the question of whether all participants are proficient in English, which could impact the survey responses.

12.  The study's reliance on health outcomes from the past 12 months may not accurately reflect the experiences of children who have lived in the area for less than a year, as their health could also be influenced by other factors, such as moving to a household with prevalent smoking

13.  The authors are encouraged to provide a clearer explanation of the multiple logistic regression models used. Specifically, it would be helpful to know if all variables were included in the model or only those that showed significant associations in single logistic regression analyses (p-value <0.05).

14.  It is useful to provide a clearer explanation of the multiple logistic regression models used. Specifically,  all variables were included in the model or only those that showed significant associations in bivariable analyse (p-value <0.05).

Results

1.     The presentation of the results seems inconsistent. For example, in Table 2, the smoking status of both father and mother, and whether or not there were smokers in the same residence were presented. As expected in the method section, these were the environmental tobacco sources and directly related to the outcome (wheezing). However, in Table 3, risk factors related to wheeze and current wheeze weren’t presented father and mother smoking status while only the father’s smoking status was presented in the current severe wheeze. Additionally, the contents in each section in Table 3 were not consistent for each health outcome. The same problem also occurred between tables 4, 5 and 6. The presentation should be improved to increase the cohesion and clarity of the results.

2.     In Table 3, the author presented the prevalence of wheeze ever, current wheeze, current severe wheeze and asthma ever for rural and urban areas. The data seems weird because odd ratios were all the same for both rural and urban areas.

3.     The manuscript should consistently use "p" to denote p-values throughout, rather than alternating between "p" and "P."

4.     The description of the table should be placed at the beginning of the paragraph that describes the results (line 232)

5.     How do parents become aware of their children's exposure to smoking at school? The variable “Child exposure to smoking at school”?

6.     In the survey question directed at female parents regarding the highest level of school completed, what does the option "Other" encompass?

7.     Given that numerous variables have missing data, how does the author plan to address this issue? Missing data can significantly impact the outcomes of multiple logistic regression models. Additionally, the manuscript should clarify the mechanisms of missing data

8.     The inclusion of 37 variables in a logistic regression model may introduce issues as multicollinearity and overfitting. Furthermore, a more complex model becomes increasingly challenging to interpret.

Discussion

1.     In lines 431-435, the author argued that children were subject to ETS in car or transport. However, as shown in the data Table 2, there were many missing values for the response of child exposure to smoking in car/transport, and this was above 50%. This could be difficult to conclude based on incomplete information in this situation. Another to take into consideration is the proportion of children walking to school which is 53%. Getting to school on a walk seems to be prevalent among preschool children. Combining these two information, was there a chance that children who also had poor health (wheezing, asthma) were too sick to walk to school and had to get to school by types of transportation? This could be supported by the fact that the authors found no association between truck traffic on the street and health outcomes.

The treatment of missing values is critical and should be addressed before it could be considered for publication.

Author Response

Dear Reviewer

Thank you for the comments.

Please see the attached responses.

Regards

Reviewer 2 Report

Comments and Suggestions for Authors

This is a study to investigate the prevalence of wheezing among preschool children residing in rural and urban areas within Mpumalanga province, and its relationship with exposure to tobacco smoke.

The findings of this article are in line with previous studies, providing relevant epidemiological information for South Africa, but does not provide relevant news for the field.

In relation to the methodology:
The study population has been established and the sample size has been calculated.
Reference is made to the inclusion of preschoolers whose parents have agreed to participate in the study.
It is not specifically reported whether they have signed an informed consent.
There is also no reference to whether any ethics committee has approved the research.

One of the questions asked presupposes prior knowledge about the existence of asthma; How was the diagnosis carried out?
In these cases, no mention is made of previous treatments with bronchodilators.

How have the variables been chosen to carry out adjusted analyses? What are the criteria for deciding between risk factors and confounders?

The wording of the text is somewhat disordered, making it difficult to compare with the information provided in the tables. Somet comparisons are made about raw  or adjusted ORs, or apparently independently analyzed data are compared: "...those children exposed to smoking at the restaurant for one or more days in the past 30 days were more likely to present with current wheeze as compared to those who lived with non-smokers."
The general impression is that  a "fishing expedition" has been made, hence the variegated data and the length of the tables.

The conclusions are consistent with previous studies and with the arguments presented.However, the data analysis and the results shown must be reviewed according to previous comments.

The paper should be supported by more recent references. 87% of them are quite old.

There is some discrepancy between the information in the text and that provided in the tables. For example, Table 1 shows 773 residents in urban areas (24.5%). However, the text lists 774 (25%).
The presentation of the information must be improved: the tables are too extensive and, at times, provide little relevant information, making them tedious to read.
It would be necessary to summarize them and modify their design.

Author Response

(The authors gave the same response as above.)

Round 2

Reviewer 1 Report

Comments and Suggestions for Authors

We have a comment about the observations and data analysis approach:

5.     How do parents become aware of their children's exposure to smoking at school? The variable “Child exposure to smoking at school”.

Author response: Parents become aware of their children's exposure to smoking at school through various means. They may learn about it from their children, who might share their experiences or observations(age-dependent). Or parents can observe teachers smoking at school.

Reviewer Comment: This indicates that some parents may not have complete awareness of their children's smoking encounters at school, as children often conceal such activities. Consequently, parents may not provide accurate information, potentially skewing the results of the study. This problem extends to questions regarding children's exposure to smoking in vehicles and restaurants, evidenced by a data missing rate of over 50%. When incorporating these variables into a multiple logistic regression model in STATA, the analysis will be based on the smallest number of observations for any variable. For instance, as shown in Table 5, the number of observations for the variables varied from 1,437 to 3,062. Hence, the multiple logistic regression model was conducted based on the 1,437 observations.

Author Response

Reviewer comment

Researcher response

How do parents become aware of their children's exposure to smoking at school? The variable “Child exposure to smoking at school”.

Author response: Parents become aware of their children's exposure to smoking at school through various means. They may learn about it from their children, who might share their experiences or observations(age-dependent). Or parents can observe teachers smoking at school.

5.1. Reviewer Comment: This indicates that some parents may not have complete awareness of their children's smoking encounters at school, as children often conceal such activities. Consequently, parents may not provide accurate information, potentially skewing the results of the study.

Thank you for the comment. As mentioned in the limitations section of our study, we addressed the issue of missing data, which could lead to variation in the totals, as indicated in the footnotes for the tables. We acknowledge that there could be various reasons for parents not responding to certain questions, which we have highlighted as a challenge in our study in the limitations section, see page 18 (lines 453-456).

5.2. This problem extends to questions regarding children's exposure to smoking in vehicles and restaurants, evidenced by a data missing rate of over 50%. When incorporating these variables into a multiple logistic regression model in STATA, the analysis will be based on the smallest number of observations for any variable. For instance, as shown in Table 5, the number of observations for the variables varied from 1,437 to 3,062. Hence, the multiple logistic regression model was conducted based on the 1,437 observations.

Thank you for the comment. Just as we have mentioned in the comment above, there could be several reasons why parents did not respond to this question. The authors have acknowledged missing data as one of the limitations of the study, see page 18 (lines 453-456). This recognition underscores the need for further investigation and perhaps a more comprehensive approach to gathering data for future research projects and ensuring or minimising the possibility of missing data.